

# A meta-analysis contrasting active versus passive restoration practices in dryland agricultural ecosystems

M. Florencia Miguel[1], H. Scott Butterfield[2] and Christopher J. Lortie[3,4]

[1] Consejo Nacional de Investigaciones Científicas y Técnicas, Mendoza, Argentina
[2] The Nature Conservancy, San Francisco, CA, USA
[3] National Center for Ecological Analysis and Synthesis (NCEAS), Santa Barbara, CA, USA
[4] Department of Biology, York University, Toronto, ON, Canada

## ABSTRACT

Restoration of agricultural drylands globally, here farmlands and grazing lands, is a priority for ecosystem function and biodiversity preservation. Natural areas in drylands are recognized as biodiversity hotspots and face continued human impacts. Global water shortages are driving increased agricultural land retirement providing the opportunity to reclaim some of these lands for natural habitat. We used meta-analysis to contrast different classes of dryland restoration practices. All interventions were categorized as active and passive for the analyses of efficacy in dryland agricultural ecosystems. We evaluated the impact of 19 specific restoration practices from 42 studies on soil, plant, animal, and general habitat targets across 16 countries, for a total of 1,427 independent observations. Passive vegetation restoration and grazing exclusion led to net positive restoration outcomes. Passive restoration practices were more variable and less effective than active restoration practices. Furthermore, passive soil restoration led to net negative restoration outcomes. Active restoration practices consistently led to positive outcomes for soil, plant, and habitat targets. Water supplementation was the most effective restoration practice. These findings suggest that active interventions are necessary and critical in most instances for dryland agricultural ecosystems likely because of severe anthropogenic pressures and concurrent environmental stressors—both past and present.

Corresponding author
M. Florencia Miguel,
fmiguel@mendoza-conicet.gob.ar

## INTRODUCTION

Dryland ecosystems are a dominant land cover type globally (*White & Nackoney, 2003*), encompassing many natural habitats such as grasslands, shrublands, and deserts (*Reid et al., 2005*). Human-influenced agricultural ecosystems such as farmlands and grazing lands are also common in drylands globally (*Ramankutty et al., 2008*; *Kennedy et al., 2019*). Natural habitats within dryland ecosystems are hotspots of biodiversity and provide important ecosystem services including food provision, water regulation, and carbon sequestration (*Millennium Ecosystems Assessment, 2005*; *Castro, Quintas-Soriano & Egoh, 2018*;

*Díaz et al., 2018*). These services support nearly 40% of the world's population (*Castro, Quintas-Soriano & Egoh, 2018*). However, land conversion, land degradation and climate change (*Reynolds et al., 2007*; *Webb et al., 2017*) have greatly impacted these ecosystems (*Millennium Ecosystems Assessment, 2005*) leading to some of the highest concentrations of threatened and endangered species worldwide (*Bonkoungou, 2001*). Drylands are thus an important set of ecosystems to manage from both anthropogenic and ecological perspectives, and synthesis of existing research will inform knowledge for balancing restoration with opportunity and change.

Dryland ecosystem degradation has led to an increase in restoration actions recognizing the high vulnerability of these systems to human impacts (*Reynolds et al., 2007*; *James et al., 2013*) and the multiple restorative benefits both to natural systems and human populations (*Clewell & Aronson, 2006*; *De Groot et al., 2013*; *Castro, Quintas-Soriano & Egoh, 2018*). Ecological restoration includes many different types of practices oriented to the recovery of degraded ecosystems and focuses on diverse targets such as plants, animals, soils, habitats, and ecosystem functions (*Clewell & Aronson, 2006*; *Perino et al., 2019*). Based on the amount of resources and human effort invested, restoration practices can be classified as active (i.e., requiring human input) or passive (i.e., requiring limited to no human input or removal of input such as perturbations) (*Hobbs & Cramer, 2008*). The class of restoration practice to implement depends on the type and extent of damage to the ecosystem. For example, the type and extent of damage can vary greatly from farmlands – where natural plant cover and animal species have been completely removed— to grazing lands—which still have a significant presence of both. Generally, more degraded ecosystems will require more active efforts to be restored (*Hobbs & Cramer, 2008*). The specific restoration goal and the availability of funding will further define the type of restoration applied locally or regionally (*Miller & Hobbs, 2007*; *Aronson & Alexander, 2013*; *Chazdon et al., 2017*). Therefore, a synthesis comparing active and passive practices will provide an analysis of trends in global restoration practices, identify outcomes reported from different restoration practices (*Gavin, 2010*), and will therefore begin to inform future restoration so as to ensure limited resources are applied using evidence and, as efficiently as possible, in a given general context of constraints.

Global agricultural intensification is likely to continue in order to meet the demands of a growing human population (*Tilman et al., 2011*; *World Health Organization, 2018*). However, global water scarcity particularly in dryland agricultural ecosystems—farmlands and grazing lands—impacted by overexploitation, land degradation, and climate change is increasingly driving retirement of those agricultural lands that are no longer productive (*Benayas et al., 2007*; *ELD Initiative, 2015*). For instance, more than 200,000 acres of irrigated farmlands in California are predicted to be retired in the next 10–20 years as part of a strategy for sustainable groundwater use (*Kelsey et al., 2018*; *Hanak et al., 2019*; *Bryant et al., 2020*). This general sociopolitical and ecological context provides the opportunity to re-claim some of these lands for native plants and animals through habitat restoration (*Queiroz et al., 2014*; *Kelsey et al., 2018*). Limitations in lands as set asides for plant and animal species has been proposed as a critical issue in all systems globally (*Diamond, 1975*; *Gotelli & McCabe, 2002*) and evidence-informed decisions for habitat

PeerJ ___________________________________________________________

restoration can contribute to the recovery of plant and animal species worldwide. Restoration of agricultural drylands back to habitat for plants and animals will provide capacity for reductions in species loss in these biodiversity hotspots (*Durant et al., 2012*; *Lortie et al., 2018*) and contribute to more secure water and food resources for a rapidly expanding human population (*Tilman et al., 2011*; *Kelsey et al., 2018*).

The main purpose of this global synthesis was to examine the extent of research on specific restoration practices in agricultural dryland ecosystems and to identify any general trends in the success of these practices for restoring native habitat. The following three goals were specifically examined: whether a classification of restoration practices into active and passive is a meaningful simplification of the complexity of restoration research for dryland agricultural ecosystems; the overall effectiveness of active vs passive restoration practices; and the outcomes identified by the restoration practices reported. The outcomes of a synthesis on drylands restoration practices can be used as a mechanism to structure evidence-based discussions and planning by researchers and stakeholders, and to yield insight into the interventionist efforts carried out for agricultural dryland restoration.

## SURVEY METHODOLOGY

### Literature search and eligibility criteria

PRISMA (Preferred Reporting Items for Systematic reviews and Meta-Analyses) guidelines were used to structure this meta-analysis (Fig. 1) (*Moher et al., 2009*). We systematically searched Scopus and The Web of Science using the following term combinations: [restoration* desert* vegetation*] OR [restoration* grassland* desert*] OR [restoration desert* plant*] OR [restoration "agricultural lands"] OR ["restoration techniques" desert*] OR ["passive restoration" desert* plant*] OR ["active restoration" desert* plant*] OR [revegetation abandoned desert*] OR [restoration "agricult*land*" desert* plant*] OR [restoration dryland* vegetation] OR [restoration semiarid* plant*] OR [restoration arid* plant*]. The searches were done in September 2018 and then updated in January 2020 and returned 2077 published articles. We collected data from studies that met the following inclusion criteria: (1) research articles that included numerical results; review and theoretical articles were not included; (2) agriculture (farmlands and grazing lands—we grouped these agricultural practices within our analysis even though there may be reasons to conduct synthesis or meta-analysis for these land uses separately, including potentially large differences in the extent of disturbance and type and intensity of restoration required to return the site to a natural state) as the main disturbance reported; (3) studies demonstrating a clear comparison of restoration practices and reference groups (i.e., intact or minimally disturbed condition) (*Wortley, Hero & Howes, 2013*); (4) reported statistical analysis and significance of treatments. We categorized all the reported terms that referred to agricultural land uses into a single term entitled "agricultural dryland ecosystems" (Fig. 2 for details on terminology for the land uses included in this meta-analysis). Agricultural land uses prior to the implementation of restoration practices included a variety of crop species and grazing systems globally (Table 1).

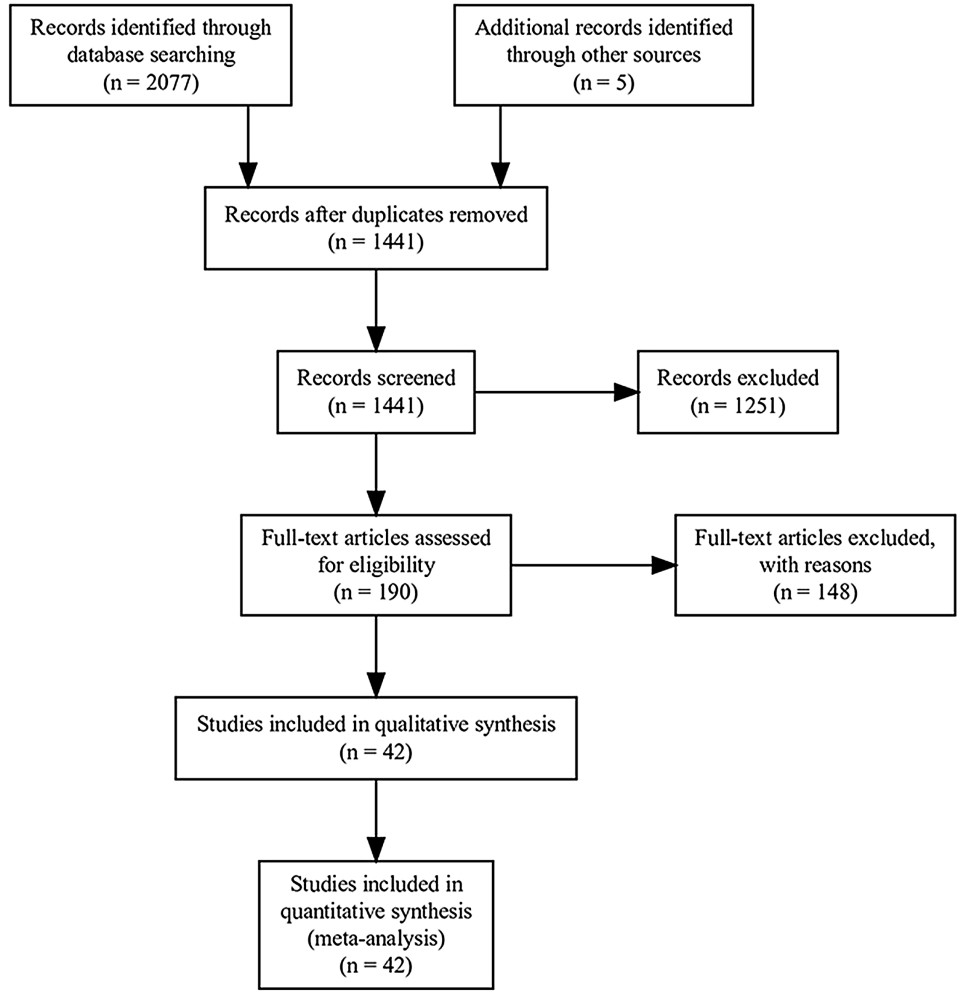

**Figure 1 PRISMA (Preferred Reporting Items for Systematic Reviews and Meta-Analyses) flowchart.** PRISMA report of a meta-analysis comparing active vs passive restoration practices in dryland agricultural ecosystems globally.

After the application of PRISMA guidelines and the above inclusion criteria, a total of 42 peer-reviewed articles (Fig. 1; Table S1) and 1,427 independent observations, from 16 different countries (Fig. 3; Table 2), were included in the meta-analysis. We defined an independent observation as a repeated, separate test in a different location listed within the 42 articles (*Koricheva, Gurevitch & Mengersen, 2013*).

## Data extraction

We extracted the following three primary elements from each article: (1) the specific restoration practice implemented (e.g., natural recovery of vegetation); (2) the restoration goal (hereafter restoration outcome) reported by the study researchers that was explicitly linked to a clearly described restoration practice (e.g., vegetation restoration); and (3) the reported response variables listed for each independent observation (Table 3).

Each of the 19 restoration practices was further categorized as active or passive to facilitate factor analysis and partition heterogeneity between studies in a transparent, a

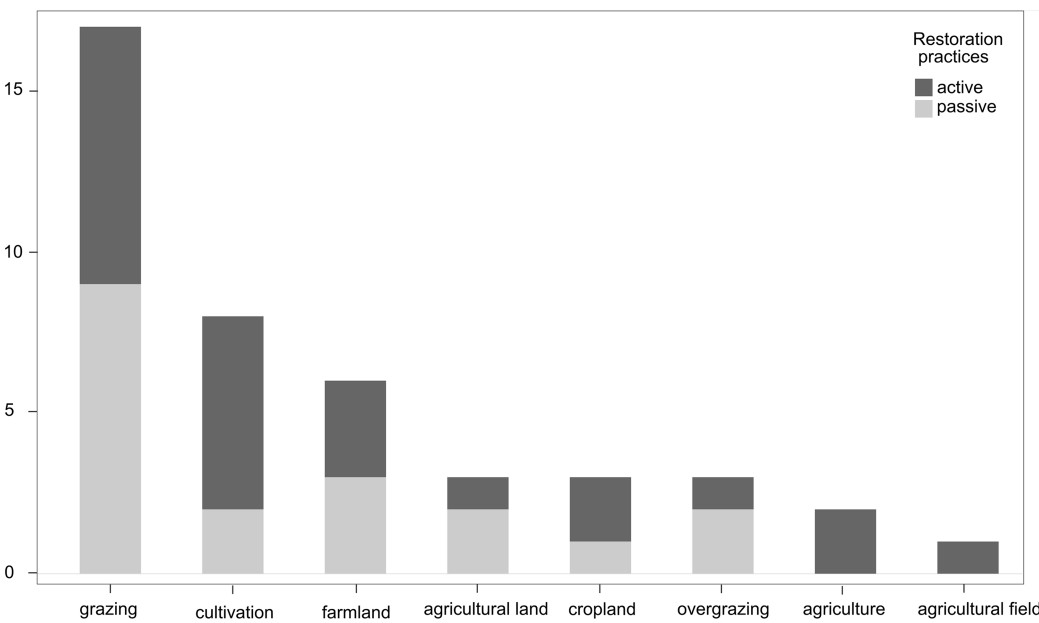

**Figure 2 Frequency of terminologies referring to agricultural land uses in dryland ecosystems.** Different terms were applied in studies included in the meta-analysis comparing active and passive restoration practices in agricultural dryland ecosystems globally. These terminologies were grouped into a single term—"agricultural dryland ecosystems", including farmlands (cultivation, farmland, agricultural land, cropland, agriculture, and agricultural field) and grazing lands (grazing and overgrazing terms).

priori designation (*Ioannidis, Patsopoulos & Evangelou, 2007*; *Koricheva & Gurevitch, 2014*). Passive restoration practices included those that focused on natural regeneration with minimal to no human interventions (*DellaSala et al., 2003*; *Hobbs & Cramer, 2008*), such as the cessation of disturbance from cattle or livestock grazing (*Filazzola et al., 2020*). Active restoration practices included those that involved direct human interventions (*Holl & Aide, 2011*), such as remediating soil (e.g., nutrient addition) and adding vegetation back to the system (e.g., planting and seeding).

Specific restoration practices were rarely replicated globally. Consequently, we classified passive restoration practices into the following categories based on their primary focus: soil, vegetation, and grazing exclusion. Passive restoration practices focused on soils restoration, such as mycorrhizal recovery, or on vegetation restoration, such as plant facilitation, were classified within soil and vegetation categories respectively (Table 3). Grazing exclusion was classified as a passive restoration practice because grazing was removed, and no other interventions were applied. We grouped active restoration practices into the following categories based on their primary focus: soil, vegetation, and water supplementation. Active restoration practices focused on soils restoration, such as mycorrhizal inoculation, or on vegetation restoration, such as seeding, and were classified within soil and vegetation categories, respectively. Water supplementation included practices such as irrigation (Table 3).

Restoration outcomes for active and passive practices were grouped into the following categories: soil, vegetation, animals, and habitat. The response variables related to soil

**Table 1 List of crop and animal grazer species in farmlands and grazing lands prior to the implementation of active and passive restoration practices in dryland agricultural ecosystems globally.** Each restoration practice was categorized as active or passive. Different practices were grouped into general categories based on their primary focus, for example those related with plant interventions such as planting or seeding, were included within the vegetation category.

| Farmlands | Grazing lands | Restoration | Category of practices | Practices |
|---|---|---|---|---|
| *Avena chinensis* | cattle | Active | vegetation | seeding |
| *Brassica nigra* | livestock | | | seeding, mowing and herbicide, mulching |
| *Erigeron canadensis* | | | | seeding, mulching, weeding |
| *Fagopyrum sagittatum* | | | | seeding, irrigation |
| *Kochia scoparia* | | | | seeding and ripping |
| *Lactuca scariola* | | | | mechanical disturbance and seeding |
| *Linum usitatissimum* | | | | seeding, safe sites for seeds and fencing |
| *Medicago sativa* | | | | planting |
| *Pisum sativum* | | | water supplementation | water supplementation |
| *Salsola iberica* | | | | seeding and irrigation |
| *Schismus spp* | | | | |
| *Sesamum indicum* | | | | |
| *Solanum tuberosum* | | | | |
| *Triticum aestivum* | | | | |
| cotton | | | | |
| corn | | | | |
| mellon | | | | |
| pecans | | | | |
| watermelon | | | | |
| not listed in studies | | | soil | carbon addition |
| | | | | mycorrhizal inoculation |
| *Medicago spp* | sheep and goat | Passive | vegetation | natural recovery |
| *Melilotus albus* | cattle | | | grazing exclusion |
| *Mentha spp* | livestock | | | facilitation |
| *Triticum aestivum* | | | | |
| *Zea mays* | | | | |
| fruits-vegetables | | | | |
| forage crops | | | | |
| – | livestock | | grazing exclusion | grazing exclusion |
| | cattle | | | facilitation |
| | sheep | | | natural recovery |
| | livestock | | | |
| *Solanum tuberosum* | – | | soil | natural recovery |
| cereal crops-fallow | | | | mycorrhizal recovery |
| chinese onion | | | | |
| peanut | | | | |
| sorghum | | | | |

measures (e.g., nutrient content) were included within the soil category. The vegetation category included plant measures such as plant cover and abundance. The animal category included measures of invertebrate abundance, diversity, and richness. The studies that met the above inclusion criteria and were related to the restoration of animal populations, were focused exclusively on invertebrates. This could be a limitation when trying to generalize about animal restoration practices and outcomes. For the vegetation and animal studies, we did not record species composition or species provenance. The habitat category
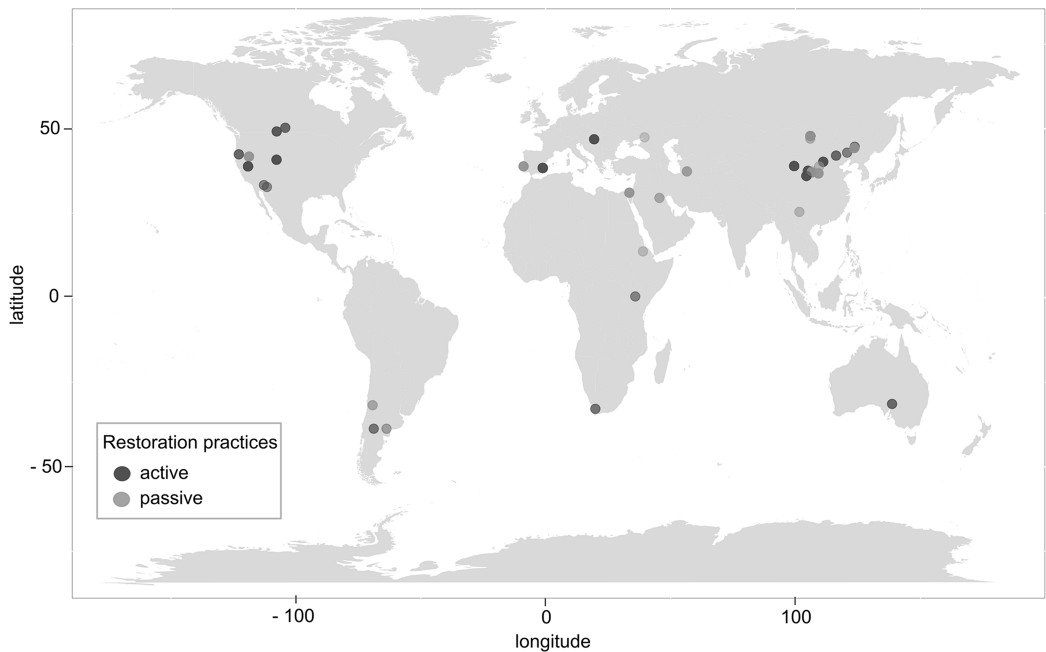

**Figure 3 Global distribution of studies evaluating restoration practices in dryland agricultural ecosystems ($n$ = 42).** Restoration practices included in the meta-analysis were classified into active or passive. Dark gray points represent the location of studies that used active restoration practices. Lighter gray points represent the location of studies that used passive restoration practices.

included measures of both soil and vegetation restoration reported together, or measures of general community structure, such as plant productivity and species evenness (Table 3).

For each reported response variable, we extracted the mean and standard deviation, the number of replicates, and the target taxa for restoration. These quantitative data were extracted for the two groups evaluated at each study including the treatment and reference groups. When these data were provided in figures within an article, we used WebPlotDigitizer (*Rohatgi, 2019*) to extract values. In addition, we collected mean annual temperature and annual precipitation data from each study site to calculate an aridity index (*Martonne, 1927*), and then recorded the reported duration of each study. When climatic data were not provided in the studies, we used the latitude and longitude listed to look up the means from WorldClim (www.worldclim.org). The aridity index and duration of studies were used as covariates in our statistical models. We also reviewed the spatial grain size (i.e., minimum size of units of observation) (*Gustafson, 1998*) of each study.

## Statistical analysis

To estimate the effect of active and passive restoration practices, we calculated the log response ratio (*lrr*) (*Hedges, Gurevitch & Curtis, 1999*). This index measures the effect size of a treatment over a control group (*Lajeunesse, 2015*); in this work, *lrr* represents the effect size of the restoration practice as the log-proportional change between the means of the treatment and reference groups. Thus, a positive *lrr* value indicates the effect of the

**Table 2 Distribution of studies evaluating restoration practices in dryland agricultural ecosystems.**
List of countries ($n = 16$) included in the meta-analysis, their active or passive restoration focus and the restoration practice implemented. Different restoration practices were grouped into general categories based on their primary focus, for example those related with plant interventions such as planting or seeding, were included within the vegetation category.

| Country | Restoration | Category of practices | Data entries |
|---|---|---|---|
| Argentina | Active | vegetation | 13 |
| | Passive | vegetation | 10 |
| Australia | Active | vegetation | 12 |
| Canada | Active | vegetation | 30 |
| China | Active | vegetation | 622 |
| | | water supplementation | 12 |
| | Passive | grazing exclusion | 30 |
| | | soil | 204 |
| | | vegetation | 42 |
| Egypt | Passive | vegetation | 21 |
| Ethiopia | Passive | grazing exclusion | 5 |
| Hungary | Active | soil | 27 |
| Iran | Passive | vegetation | 15 |
| Kenya | Active | vegetation | 7 |
| Kuwait | Passive | vegetation | 10 |
| Mongolia | Active | vegetation | 24 |
| | Passive | vegetation | 37 |
| Portugal | Passive | grazing exclusion | 21 |
| Russia | Passive | soil | 4 |
| South Africa | Active | vegetation | 9 |
| Spain | Active | soil | 128 |
| United States of America | Active | vegetation | 57 |
| | | water supplementation | 63 |
| | Passive | vegetation | 24 |

restoration practice was higher than that of the reference group (i.e., the restoration practice has a positive impact on restoration outcomes) while a negative *lrr* value indicates the effect of the reference or control group was higher than that of the restoration practice. A *lrr* value of zero represents no net effect of the restoration practice on restoration outcomes (*Pustejovsky, 2018*).

We used random effects models to account for the variability between studies (e.g., different restoration practices implemented, response outcomes pursued, and response variables measured) (*Schwarzer, Carpenter & Rücker, 2015*). Post hoc meta-regressions were then used to test the influence of aridity (*Martonne, 1927*) and time from onset of study. Statistical significance of active and passive restoration practices was tested with *t*-tests against a value of 0. Restoration practices and outcomes were considered significant if their estimated 95% confidence intervals did not overlap 0 (*Cote & Jennions, 2013*). All analyses were done in R version 3.4.4 (*R Core Team, 2018*). The meta and metaphor packages were used for the meta-analysis (*Schwarzer, 2007*; *Viechtbauer, 2010*).

**Table 3 List of restoration practices, desired restoration goals (i.e., outcomes) and original response variables included in the meta-analysis.** Data was used to compare active vs passive restoration practices in dryland agricultural ecosystems globally ($n$ = 42 and 1,427 independent observations or data entries). Different practices were grouped into general categories based on their primary focus, for example, those related with plant interventions such as planting or seeding, were included within the vegetation category. The outcomes listed describe restoration goals from each restoration practice; the habitat classification includes studies that reported measures of both soil and vegetation recovery or of vegetation community structure.

| Restoration | Category of practices | Practices | Outcomes | Response variables | Data entries |
|---|---|---|---|---|---|
| Active | soil | carbon amendment | soil | moss cover; soil nutrient content | 27 |
| | | mycorrhizal inoculation | vegetation | plant biomass; nutrient | 128 |
| | vegetation | burning, mowing | habitat | soil nutrient content and soil properties | 24 |
| | | mechanical disturbance, seeding | vegetation | plant cover and density | 4 |
| | | planting | habitat | plant biomass, density, cover, diversity and richness; soil nutrient | 369 |
| | | planting | vegetation | plant height and cover; invertebrate and lizard abundance, diversity, dominance, evenness and richness | 26 |
| | | planting | soil | soil nutrient content and soil properties | 84 |
| | | planting | animals | invertebrate abundance, diversity and richness | 6 |
| | | planting, grazing exclusion | animals | arthropod abundance, richness and diversity; soil properties; plant cover, density, height and richness | 24 |
| | | seeding | vegetation | plant cover and density; seedling emergence and establishment | 53 |
| | | seeding | soil | soil nutrient content and soil properties | 117 |
| | | seeding and ripping | vegetation | plant cover and abundance | 12 |
| | | seeding, gypsum and organic mulch | habitat | soil properties; seedling emergence and surviving plants | 9 |
| | | seeding, irrigation | vegetation | seedling emergence | 7 |
| | | seeding, mowing and herbicide, mulching | habitat | plant cover and richness; soil nutrient content and soil properties | 18 |
| | | seeding, mulching, weeding | vegetation | plant cover | 6 |
| | | seeding, safe sites for seeds, fencing | vegetation | plant cover and biomass | 8 |
| | | seeding, soil tilling, fertilization | vegetation | plant biomass | 7 |
| | water supplementation | irrigation, seeding | vegetation | plant cover, abundance, biomass, density and survival | 63 |
| | | water supply | habitat | plant biomass, density, cover, evenness, productivity and richness; soil nutrient content | 12 |
| Passive | grazing exclusion | fencing | vegetation | plant height, cover and diversity | 21 |
| | | grazing exclusion | vegetation | plant height, cover, diversity, biomass and richness | 8 |
| | | natural recovery | vegetation | plant biomass, cover, density, height | 27 |
| | soil | mycorrhizal recovery | soil | microbial richness and density | 6 |
| | | natural recovery | soil | soil nutrient content and soil properties | 202 |
| | vegetation | facilitation | habitat | soil nutrient content and soil properties; plant survival, biomass, height, width, abundance, and richness | 60 |
| | | natural recovery | habitat | soil nutrient content and soil properties; plant richness | 40 |
| | | natural recovery | animals | arthropod density, diversity and richness | 3 |
| | | natural recovery | soil | soil properties | 18 |
| | | fencing | habitat | plant biomass, evenness, cover, density, diversity, height and richness; soil nutrient content and soil properties | 15 |
| | | grazing exclusion | vegetation | plant cover, density, height, biomass and richness | 23 |

**Table 4 The effect of active and passive restoration practices and restoration outcomes evaluated in dryland agricultural—here, defined as farmlands and grazing lands—ecosystems globally.** The log response ratio (effect size) and 95% confidence interval (CI) were calculated from random effects models (Lortie, C.J. and Miguel, M.F. 2019. R code, DOI 10.5281/zenodo.3907012). Effect of active and passive restoration practices was tested by *t*-test against a value of 0, and restoration practices and outcomes were considered significant if their estimated 95% confidence intervals did not overlap 0. (A) Random effects model results comparing restoration practices. (B) Random effects model results comparing restoration outcomes. Outcomes describe target goals from each restoration practice; the habitat category includes studies that reported measures of both soil and vegetation restoration or general community structure.

| Restoration | Log response ratio | 95% CI |
|---|---|---|
| (A) | | |
| Active practices | 0.34 | [0.27–0.42] |
| Water supplementation | 0.64 | [0.55–0.73] |
| Soil | 0.56 | [0.54–0.57] |
| Vegetation | 0.19 | [0.18–0.21] |
| Passive practices | −0.29 | [−0.36 to −0.21] |
| Soil | −0.74 | [−0.81 to −0.68] |
| Vegetation | 0.23 | [0.18–0.28] |
| Grazing exclusion | 0.13 | [0.06–0.20] |
| (B) | | |
| Active restoration outcomes | | |
| Vegetation | 0.50 | [0.49–0.52] |
| Soil | 0.28 | [0.21–0.35] |
| Habitat | 0.10 | [0.09–0.12] |
| Animals | −0.11 | [−0.113 to −0.112] |
| Passive restoration outcomes | | |
| Soil | −0.68 | [−0.74 to −0.62] |
| Vegetation | 0.29 | [0.23–0.35] |
| Habitat | 0.13 | [0.07–0.19] |
| Animals | 1.05 | [−0.21 to 2.31] |

## RESULTS

This meta-analysis included 42 peer-reviewed articles covering 16 countries in dryland agricultural ecosystems (Fig. 3; Table 2). The meta-analysis included the evaluation of 19 different restoration practices, categorized into three active and three passive practices, on restoration outcomes (Table 3). There were a total 1,427 independent observations (or data entries) from the 42 articles that were analyzed in the meta-analysis (*Miguel, Butterfield & Lortie, 2020*). The mean spatial grain size for the studies was 2,320.1 $m^2$ for active and 814.15 $m^2$ for passive restoration practices (Table S2).

Active restoration practices consistently led to positive restoration outcomes (Table 4). All three categories of active restoration, including soil, vegetation, and water supplementation, had net positive responses. Water supplementation was the most effective restoration practice, followed by soil and vegetation restoration practices (Table 4A; Fig. 4). When analyzing restoration outcomes, we found that soils, vegetation,

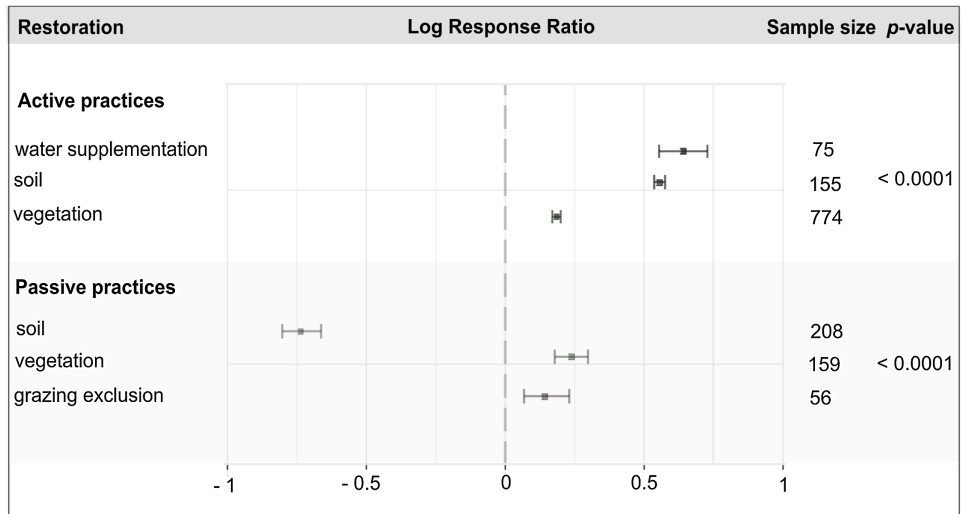

**Figure 4 Log response ratio (effect size) and 95% confidence intervals for active and passive restoration practices in agricultural dryland ecosystems.** The dashed vertical line denotes no effect of restoration practices, or a mean of zero. A positive log response ratio value indicates the mean of the restoration practice was higher than that of the reference condition and a negative value indicates the mean of the reference condition was higher than that of the restoration practice. The p-values are from random effect models comparing subgroups differences among restoration practices.

and habitat are likely to be restored through active restoration practices, but that invertebrate animal communities were not (Table 4B).

Passive restoration practices had lower and more variable effect sizes when compared to active restoration practices (Table 4A). Passive restoration of vegetation and grazing exclusion had positive effects on restoration outcomes (Table 4A; Fig. 4). However, passive soil restoration practices led to negative restoration outcomes (Table 4A; Fig. 4). Soils did not passively recover in agricultural drylands, but plants and habitat did passively recover in some instances (Table 4B).

Aridity had a weak negative impact on active restoration practices suggesting that water limitations can mediate the positive results of these practices on restoration outcomes (*lrr* aridity = −0.02, 95% CI [−0.03 to −0.02]). For active restoration practices, increasing the duration of the study had a significant, but small positive impact on restoration (*lrr* time = 0.003, 95% CI [0.002–0.0034]), suggesting longer studies be considered when evaluating the impact of active restoration practices on restoration outcomes. For passive restoration practices, the duration of the study negatively influenced restoration outcomes, but variation in aridity had no effect (*lrr* time = −0.003, 95% CI [−0.003 to −0.002]; *lrr* aridity = −0.003, 95% CI [−0.006 to 0.005]).

## DISCUSSION

### The need for a meta-analysis of restoration practices in dryland agricultural ecosystems

Active vs passive restoration strategies is a critical decision in the management of agricultural drylands globally, and the aggregated evidence confirmed that there

are consistent and crucial differences between this simple grouping of practices. The opportunity to restore agricultural systems is increasing globally particularly in farmlands in dryland ecosystems that are experiencing intensifying water shortages and resulting land retirement (*Benayas et al., 2007*; *Kelsey et al., 2018*). Unfortunately, ecological restoration is neither a simple concept nor "one size fits all" group of practices with known outcomes (*Higgs, 1997*). Instead, there are numerous potential combinations of restoration practices across most systems, including in drylands, which are infrequently replicated globally. The low replicability of individual restoration practices is a potential limitation for using the results of this synthesis to inform global decision making for restoration. However, synthesis of practices helps to simplify and aggregate the global evidence to explore generality and to advance theory in this field.

Ecological restoration is a broad set of interventions that comprise practices conducted in a wide range of ecosystems globally (*Hobbs & Cramer, 2008*). In tropical and temperate rain forests, previous meta-analyses have shown that passive restoration including natural succession processes can lead to positive, desired plant and animal restoration outcomes (*Crouzeilles et al., 2017*; *Meli et al., 2017*). Nevertheless, for drylands, this meta-analysis showed that active restoration practices more consistently led to positive restoration outcomes. The likelihood of efficacy between active and passive restoration practices can be explained by the physical constraints of these ecosystems that experience relatively severe limitations in rainfall, soil fertility, and productivity (*Millennium Ecosystems Assessment, 2005*). To this end, aridity was a significantly limiting factor in the models for active restoration outcomes suggesting that drylands pose unique challenges and considerations for effective application of interventions. Moreover, the extent of land transformation and the type of prior land use can also contribute to the requirement of increased efforts and investments (*Holl & Aide, 2011*) to achieve agricultural dryland restoration. Collectively, this evidence supports previous research and highlights the need for consideration of environmental limitations in drylands.

## The outcome of active vs passive restoration practices

Importantly, active restoration practices are required to achieve soil-based outcomes in farmlands in agricultural drylands, while passive practices lead to negative soil restoration outcomes. Because soils constitute the foundation for long-term ecosystem recovery (*Costantini et al., 2016*), it is likely that any successful effort would require some form of active restoration. For instance, mycorrhizal inoculation contributes to the restoration of soil microorganisms and the subsequent successful establishment and growth of the desired shrub species (*Caravaca et al., 2003*); and carbon addition increases the availability of soil nutrients for plants and moss cover in former agricultural drylands (*Török et al., 2014*). This result is consistent with previous work from the San Joaquin Desert of California that recommends any restoration project on formerly farmed lands start with soil nutrient remediation (*Laymon et al., 2010*). Despite the fact that we did not separately evaluate restoration outcomes for farmlands and grazing lands in this study, soil restoration efforts are much more likely to be required in farmlands (vs grazing lands)

because of the extent of the damage – including from tilling and synthetic inputs to increase crop productivity—in these intensively managed systems (*Garibaldi et al., 2019*; *Kleijn et al., 2019*). Although resources can be limiting for restoration, particularly for large-scale projects that will have the most significant impacts on ecosystem services and biodiversity, active restoration may be necessary in order to overcome the legacies of soil disturbances, nutrient additions, and pesticide usage (*Kleijn et al., 2019*) in agricultural drylands.

Once soil restoration is achieved, plant restoration can proceed, actively or passively. The passive restoration of plant species is an emerging strategy for restoring native communities with minimal costs (*Hobbs & Cramer, 2008*; *Tabeni et al., 2017*). Moreover, the removal of grazing was an effective strategy for passive restoration in drylands, similar to the findings from a recent global grazing meta-analysis (*Filazzola et al., 2020*). Nevertheless, in some more mesic grazing systems, like in the coastal and northern interior portions of California, removing grazing can lead to greater dominance of non-native plant species and overall lower levels of native plant and animal biodiversity (*Hayes & Holl, 2003*; *Marty, 2005*). Active plant restoration such as seeding and planting also led to positive outcomes and likely requires water supplementation. However, as the species origin of the restored plant communities was not evaluated in this meta-analysis, the decision for active or passive plant restoration practices will depend on the biotic context of the site and the species-specific restoration goals of the project. Future studies can examine this limitation by specifically assessing species diversity of restored agricultural drylands under different restoration practices.

Although some passive restoration practices led to positive restoration outcomes, these results were more variable and at lower levels than the ones found by active restoration practices. The aridity of sites and the duration of treatments had contrasting influence on the restoration outcomes of different practices reflecting the dependance on context for the outcomes of restoration projects (*Gravuer, Gennet & Throop, 2018*) and the influence of physical constraints to the success of restoration practices (*Miller & Hobbs, 2007*). However, a focus on intensively (farmlands) and extensively (grazing lands) managed agricultural drylands and their restoration outcomes contributes to a more general understanding of the restoration practices because of the relatively high variety of intervention tested. Finally, data on animal restoration outcomes was limited to invertebrate community-based studies, a taxa that is known to be severely impacted by agricultural practices (*Potts et al., 2010*; *Garibaldi et al., 2011*; *Sánchez-Bayo & Wyckhuys, 2019*). This research gap highlights the difficulties in restoring animal community targets even if soil and plant restoration is successful at a specific site. More comprehensive studies of restoration outcomes and extended biodiversity analyses in agricultural drylands are necessary to better assess the extent of intervention needed.

## CONCLUSIONS

Meta-analysis is a synthesis tool particularly valuable to identify large-scale patterns and to inform evidence-based decision making for stakeholders (*Gavin, 2010*). Considering

the opportunity to restore agricultural dryland ecosystems globally, meta-analysis can reveal broad trends in data that inform decision-making about the restoration practices most likely to achieve certain restoration outcomes. This meta-analysis revealed that you can get some restoration outcomes for "free" but, as we noted, these outcomes may be more variable in these systems. Nevertheless, with limited resources, active restoration practices are required to achieve positive restoration outcomes, likely because of severe anthropogenic pressures and concurrent environmental stressors – both past and present – in dryland agricultural ecosystems. Facing the opportunity of reclaiming drylands formerly used for agricultural practices, soils will require active restoration interventions; and when resources for restoration do exist, we may focus efforts on native plant species restoration and potentially more specifically those that serve a foundational or keystone role (e.g., shrubs) (*Lortie, Filazzola & Sotomayor, 2016*) within the ecosystem.

### Funding

This research was funded by The Nature Conservancy and York University. Christopher J. Lortie was also supported as Senior Research Fellow at NCEAS and by an NSERC DG Grant in Canada. M. Florencia Miguel was also supported by a post-doctoral fellowship from CONICET. There was no additional external funding received for this study. The funders had no role in study design, data collection and analysis, decision to publish, or preparation of the manuscript.

### Grant Disclosures

The following grant information was disclosed by the authors:
The Nature Conservancy and York University.
NSERC DG Grant in Canada.
CONICET.

### Competing Interests

The authors declare that they have no competing interests. Christopher J. Lortie is an Academic Editor for PeerJ.

### Author Contributions

- M. Florencia Miguel conceived and designed the experiments, performed the experiments, analyzed the data, prepared figures and/or tables, authored or reviewed drafts of the paper, and approved the final draft.
- H. Scott Butterfield conceived and designed the experiments, authored or reviewed drafts of the paper, acquired the financial support for the project, and approved the final draft.
- Christopher J. Lortie conceived and designed the experiments, analyzed the data, authored or reviewed drafts of the paper, acquired the financial support for the project, and approved the final draft.

## Data Availability

The code is available at Zenodo: Lortie, C.J. and Miguel, M.F. 2019. A set of R code to test dryland restoration efficacy using meta-analysis. Zenodo. V2.2. DOI 10.5281/zenodo.3907012.

The data is available at Figshare: Miguel, Florencia M.; Butterfield, H. Scott; Lortie, Christopher (2020): A synthesis dataset describing dryland agricultural restoration practices. figshare. Dataset. https://figshare.com/articles/dataset/A_synthesis_dataset_ describing_dryland_agricultural_restoration_practices/12560240.

## Supplemental Information

Supplemental information for this article can be found online at http://dx.doi.org/10.7717/ peerj.10428#supplemental-information.

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
