# Peer review of "A meta-analysis contrasting active versus passive restoration practices in dryland agricultural ecosystems"

_PeerJ, doi:10.7717/peerj.10428_

## Round 0.1 · original submission · Minor Revisions

Although this is a good paper. I agree with both of the reviewers that a bit more clarification needs to be added. I think the paper could be much improved by adding examples, but also clearly stating what was included and not included in the study, and why.

·

Basic reporting

no comment

Experimental design

no comment

Validity of the findings

no comment

Additional comments

This is a global meta-analysis and synthesis of research on restoration practices in agricultural dryland ecosystems. It looked at the outcomes and effectiveness of active vs passive restoration practices. The meta-analysis resulted in 42 peer-reviewed articles containing 1,427 independent observations (or data entries) from which the restoration practice, goal, and response for each practice was assembled. Passive restoration focused on soil, vegetation , and grazing exclusion, while active restoration focused on soil, vegetation and water supplementation.

The paper is clearly written and the results are important and compelling. I just have just one suggestion for improvement:

A bit more explanation and contextualization of the results would help. For example, it would be good to add a few specific examples from the actual studies to highlight the restoration effects that are summarized in the meta-analysis results. Give a case study or two that shows how and why soil restoration was done and how it produced the desired effects.

Reviewer 2 ·

Basic reporting

No comment

Experimental design

No comment

Validity of the findings

No comment

Additional comments

The work is to a high standard, but perhaps missing some small details to add some depth to the analysis.

Results section: A table listing the countries (to complement the map) and No. of cases per country and categories of practice per country would be useful.

A paragraph explicitly acknowledging the limitations of the study and the methodology would be welcome at some point in the text.

L 303 – 305 (Word doc) This needs to be stated much sooner, if biodiversity is not considered and instead only abundance. As you state the importance of drylands ecologically in the introduction, and it is not mentioned in the methods that biodiversity was not considered.

L 51 – 52 (Word doc) It is important here to contextualise between intensive agricultural activities and extensive/traditional agricultural activities in dry lands as causes of degredation

L88 – 91 (Word doc) While agricultural intensification is responsible for degradation, traditional livestock systems can be used to increase biodiversity and the exclusion of livestock causes a decrease in biodiversity and an increase in abundance of a few species. Please try to contextualize the difference between traditional and intensive agricultural land use.

L 91 – 93 (Word doc) Please mention here if this land is to be retired because it is unproductive or if there are other factors socio-economic factors such as an aging population and lack of succession options. Due to the previous sentence it reads like this land in California is going to be retired exclusively because they are no longer productive.

L136 (Word doc) These categories represent your goals well, but I think differentiating those systems that could be considered intensive vs traditional or extensive would add depth to the analysis. Equally a list of the crops or species uses in the systems prior to restoration activities could help contextualize your study.

---

## Round 0.2 · accepted · Accept

Thank you for making the revisions to this article. Your edits have clarified the paper significantly. It's a great article.